# Differentiation of *Aspergillus flavus* from *Aspergillus oryzae* Targeting the *cyp51A* Gene

**DOI:** 10.3390/pathogens10101279

**Published:** 2021-10-04

**Authors:** Sanaz Nargesi, Mahdi Abastabar, Reza Valadan, Sabah Mayahi, Jung-Ho Youn, Mohammad Taghi Hedayati, Seyedmojtaba Seyedmousavi

**Affiliations:** 1Department of Medical Mycology, School of Medicine, Mazandaran University of Medical Sciences, Sari 48157-33971, Iran; nargesi.sanaz@gmail.com (S.N.); sabah.mayahi@gmail.com (S.M.); 2Invasive Fungi Research Center, Communicable Diseases Institute, Mazandaran University of Medical Sciences, Sari 48157-33971, Iran; 3Molecular and Cell Biology Research Center (MCBRC), Mazandaran University of Medical Sciences, Sari 48157-33971, Iran; valadan.reza@gmail.com; 4Clinical Center, Microbiology Service, Department of Laboratory Medicine, National Institutes of Health, Bethesda, MD 20892, USA; jung-ho.youn@nih.gov

**Keywords:** *Aspergillus* section *Flavi*, MALDI-TOF MS, PCR sequencing, genetic marker, phylogenetic diversity

## Abstract

*Aspergillus flavus* is one of the most important agents of invasive and non-invasive aspergillosis, especially in tropical and subtropical regions of the world, including Iran. *Aspergillus oryzae* is closely related to *A. flavus*, and it is known for its economic importance in traditional fermentation industries. Reports of infection due to *A. oryzae* are scarce. Several studies reported that differentiating these two species in clinical laboratories is not possible using MALDI-TOF or by targeting fungal barcode genes, such as Internal Transcribed Spacer (ITS) and β-tubulin (*benA*). The species-level identification of causative agents and the determination of antifungal susceptibility patterns can play significant roles in the outcome of aspergillosis. Here, we aimed to investigate the discriminatory potential of *cyp51A* PCR-sequencing versus that of the ITS, *benA* and calmodulin (*CaM*) genes for the differentiation of *A. flavus* from *A. oryzae*. In a prospective study investigating the molecular epidemiology of *A. flavus* in Iran between 2008 and 2018, out of 200 clinical isolates of *A. flavus*, 10 isolates showed >99% similarity to both *A. flavus* and *A. oryzae*. Overall, the ITS, β-tubulin and *CaM* genes did not fulfil the criteria for differentiating these 10 isolates. However, the *cyp51A* gene showed promising results, which warrants further studies using a larger set of isolates from more diverse epidemiological regions of the world.

## 1. Introduction

*Aspergillus* section *Flavi* comprises 33 phylogenetically distinct species, the majority of which are natural producers of aflatoxins and ochratoxins; among them, *A. flavus, Aspergillus tamarri, Aspergillus sojae, Aspergillus parasiticus* and *Aspergillus nomius* are mainly isolated from clinical sources [1]. In different sections of *Aspergillus*, including section *Flavi*, there are several cryptic species that, while they may not be distinguishable morphologically, are quite unique in their pathogenesis profile, as well as their patterns of susceptibility to currently available systemic antifungals [2,3]. *A. flavus* is the most well-known species in section *Flavi* and a leading cause of severe invasive and non-invasive fungal infections in the Middle Eastern region [4]. The differentiation of *A. flavus* from *A. oryzae* is a significant challenge in clinical mycology laboratories, which use conventional mycology, focusing on macroscopic and microscopic characteristics [5], and currently available laboratory-developed tests and commercial proteomic and molecular assays. We previously showed that the Bruker Daltonik matrix-assisted laser desorption/ionization time-of-flight mass spectrometry (MALDI-TOF MS) database also cannot properly separate *A. flavus* from *A. oryzae* [6]. Recent data from whole-genome sequencing studies, however, shed new light on the genome sequences of these species, indicating homogeneity in the genome size of two species (*A. flavus*: 36.8 Mb, *A. oryzae*: 36.7 Mb), as well as the presence of the coding genes of many secondary metabolites, such as polyketide synthases and P450 enzymes, with nearly equal numbers [7]. In a computerized analysis of different gene loci, including the *Cyp51A* sequence, we found nucleotide variations in several locations, which suggested the potential for differentiation between two closely related species, *A. flavus* and *A. oryzae*. Therefore, in the current study, we aimed to study the utility of *cyp51A* in the phylogenetic differentiation of *A. flavus* from *A. oryzae*.

## 2. Material and Methods

### 2.1. Fungal Isolates

In a prospective countrywide molecular epidemiology study between 2008 and 2018 performed at the Invasive Fungi Research Center, Mazandaran University of Medical Sciences, Sari, Iran, out of 200 *A. flavus*, 10 isolates showed >99% similarity to both *A. flavus* and *A. oryzae* after PCR-sequencing of the *benA* gene. These 10 isolates were taken from the sputum or bronchoalveolar lavage fluid (BAL) of patients from different regions of Iran and with different underlying conditions, including hematological malignancies, organ transplantation, chronic obstructive pulmonary disease, and lung cancer. These 10 isolates were subject to further work to discriminate between the two mentioned strains based on the PCR sequencing of the *benA*, ITS, *CaM* and *cyp51A* genes using the NCBI GenBank (http://www.ncbi.nlm.nih.gov/genbank, accessed on 14 January 2021) and the International Mycological Association-Westerdijk Fungal Biodiversity Institute (http://www.mycobank.org, accessed on 14 January 2021) databases. The corresponding sequences were submitted to GenBank. Additionally, two standard species, *A. oryzae* (RIB40) and *A. flavus* (NRRL3357), were also used to characterize the *cyp51A* gene sequences. All the isolates were stored at −80 °C in glycerol broth and cultured on potato dextrose agar and/or Sabouraud’s dextrose agar (SDA) supplemented with chloramphenicol for 5–7 days at 35 °C to ensure purity and good sporulation.

### 2.2. Molecular Identification

#### 2.2.1. DNA Extraction

The total DNA of all the fungal strains was extracted from the surface harvest of colonies cultured on SDA using the methodology described previously (15). A total of 50 μL of sterile distilled water was added and the samples were preserved at −20 °C.

#### 2.2.2. PCR Amplification and Sequences Analysis of the ITS, *benA*, *CaM*, and *cyp51A* Genes

The extracted DNA was used as a template for the amplification of the ITS region of ribosomal DNA using ITS1 primer (5′-CTTGGTCATTTAGAGG AAGTAA-3′) and ITS4 primer (5′-GGTCCGTGTTTCAAGACGG-3′); the *benA* genewas amplified using the BT2a (5′-GGTAACCAAATCGGTGCTGCTTTC-3′) and BT2b (5′-ACCCTCAGTGTAGTGACCCTTGGC-3′) primers; the *CaM* genewas amplified using the CF1M primer (5-AGGCCGAYTCTYTGACYGA) and CF4 primer (5-TTTYTGCATCATRAGYTGGAC); and, finally, the coding sequences of the *cyp51A* gene were amplified using forward (5′-CCAGATTAGGCATACACATTG-3′) and reverse (5′-CGCTAACTATGGTTGACTCTA-3′) (1), respectively. The PCR reactions were carried out in 25 µL containing 1 µL of 25 pmol of each primer, 2.5 U of Taq DNA polymerase, 2.5 mM MgCl_2_, 400 mM deoxynucleotide triphosphates, 2 µL genomic DNA, and water. Table 1 demonstrates the PCR cycling steps for the four genes.

The *benA* and *CaM* PCR products were sequenced using only one primer. The *cyp51A* gene and ITS region PCR products were sequenced from both directions using an automated DNA Sequencer (ABI PRISM_ABI-3730 Genetic Analyzer; PE Applied Biosystem, Foster City, CA, USA). The sequences of the *cyp51A* gene were assembled and edited with a DNA Sequence Assembler (version 5.15) (http://www.dnabaser.com/download/DNA-Baser-sequence assembler/how%20to%20 download.html, accessed on 14 January 2021) and the CLC Genomics Workbench software package (CLC Bio, Qiagen, Aarhus, Denmark), aligned with reference strains and compared with each other; finally, a phylogenetic tree was constructed with the UPGMA method using MEGA software (version 10.1) (https://www.megasoftware.net/, accessed on 14 January 2021) and 1000 bootstraps. The comparative sequence analyses were performed by using the Basic Local Alignment Search Tool (BLAST) on the National Center for Biotechnology Information (NCBI) website.

## 3. Results

### 3.1. Comparison of ITS Region, benA, CaM and cyp51A Genes

#### 3.1.1. ITS Region

The ITS region of total isolates was successfully amplified by universal primer pairs (ITS1 and ITS4) and generated PCR products with a size of 1200 bp. Bioinformatics analysis, which was performed by comparing the sequence results of the ITS region with sequence data registered in the NCBI nucleotide Blast (https://blast.ncbi.nlm.nih.gov/Blast.cgi, accessed on 14 January 2021), indicated that the ITS identified both species (*A. flavus* and *A. oryzae*) equally (similarity index: 99.55%). As a result, the ITS region could not properly distinguish these two species. Therefore, all 10 isolates were reported as *A. flavus/A. oryzae* (Table 2).

#### 3.1.2. β-Tubulin Gene

The β-tubulin genes of all the isolates were partially amplified by universal primer pairs (Bt2a and Bt2b) and created PCR products with a size of 550 bp. The data obtained from the sequences provided consistent identification at the species level of only two out of the ten isolates tested. The two isolates with a 100% similarity rate were identified as *A. flavus* and the others were identified as *A. flavus/A. oryzae* (Table 2).

#### 3.1.3. *CaM* Gene

The calmodulin genes of the tested isolates were amplified by two primers, CF1M and CF4, and created products with a size of 560 bp. The *CaM* gene correctly identified three isolates as *A. flavus*, with similarity rates of 100%, 99.28% and 99.61%, respectively. Two isolates were identified as *A. flavus/A. oryzae*. The remaining *A. oryzae* isolates (*n* = 5, accession Nos. MZ291480, MZ291483, MZ291487, MZ291492, and MZ291493) were misidentified as *A. flavus* (Table 2).

#### 3.1.4. *cyp51A* Gene

The coding sequences of the *cyp51A* genes of all the strains were amplified by a size of 1542 bp (Figure 1A); the sequences were submitted to the GenBank and registered as accession numbers (MW558954–MW558963) (Table 2). The comparison of nucleotide sequences with sequences registered in the GenBank as standard strains of *A. flavus* (NW_2477238) and *A. oryzae* (RIB40), revealed that the *cyp51A* gene could assist us in separating the species-level identifications of these 10 isolates. The alignment of the nucleotide sequences of the *cyp51A* gene from the 10 tested isolates with standard strains showed four regions with nucleotide variations, located in 613, 1083, 1386 and 1389 (Figure 1A). In the *cyp51A* gene, the *A. flavus* isolates (NW_2477238, MW558954, MW558955, MW558956, MW558957, and MW558958) harbored four single polymorphisms (SNPs) at position 613 (G), 1083 (C) 1386 (C), and 1389 (G), whereas all the strains with SNPs at position 613 (A), 1083 (T), 1386 (T), and 1389 (A) were related to the *A. oryzae* isolates (RIB40, P2, P5, P8, P11 and p16) (Figure 1A). Accordingly, five isolates that were not separated with *benA*, *CaM* and ITS markers were classified as *A. flavus* (accession Nos. MW558954, MW558955, MW558956, MW558957, and MW558958), resulting in a 100% similarity to the *A. flavus* standard strain (NW_2477238), while the remaining isolates showed a 100% similarity to the *A. oryzae* standard strain (RIB40) and were categorized as *A. oryzae* (MW558959, MW558960, MW558961, MW558962, and MW558963) (Table 2). The predicted cDNA (10542 bp) of the *cyp51A* gene encoded a 514-amino acid protein, in which one nonsynonymous mutation at position 205 was detected (Figure 1B). All the *A. flavus* isolates harbored Alanine (A) in this position, while Threonine (T) was located in *A. oryzae* (Figure 1B).

### 3.2. Phylogenetic Diversity

The phylogenetic analysis derived from the *cyp51A* gene sequences constructed using the UPGMA method revealed two clades. Clade 1 clearly represented all the *A. oryzae* isolates (MW558959–MW558963) adjacent to the reference strain (RIB40) and clade 2 markedly consisted of all the *A. flavus* isolates (MW558954–MW558958) next to the standard strain (NRRL3357) (Figure 2).

## 4. Discussion

Given the worldwide increase in IA cases [8], it is crucial to identify the causative agent of disease at species level as these species may tend towards a certain geographical location or climate. Overall, different species of *Aspergillus* have unique antifungal susceptibility patterns, which is essential for recommending the appropriate antifungal treatment. *Aspergillus fumigatus* is the principal agent of IA in the most parts of the world; however, it was recently concluded that *A. flavus* has the potential to become the leading agent of IA in tropical regions, as it can adapt to climatic conditions in Asia and Africa [9]. According to a recent systematic review and meta-analysis, *A. flavus* was introduced as the most common species isolated from IA patients in Iran [10]. *A. oryzae*, on the other hand, is widely used in traditional fermentation industries but has significant similarities to *A. flavus*, both morphologically and in its ITS and *benA* gene profiles [5]. The role of *A. oryzae* in human infections was reported only as the causative agent of aspergillosis in a few cases [11,12]. Recent studies showed homogeneity in the whole-genome size of the two species and also revealed the presence of the coding genes of many secondary metabolites, such as polyketide synthases and p450 enzymes, with nearly equal numbers [7]. Polyphasic approaches were recommended as the gold standard for the classification of *Aspergillus* species because the reliance on a singular morphology or molecular-based identification methods has largely failed to identify closely related species [13,14]. Despite significant efforts and the use of a variety of techniques, including phenotypic characteristics, molecular approaches and, most recently, MALDI-TOF, the separation of *A. flavus* and *A. oryzae* is yet to be achieved successfully. Multi-locus sequencing was presented as an efficient tool to properly understand the phylogenetic relations among *Aspergillus* species [15]. The evaluation of findings based on ITS regions sequencing, known as the standard barcoding target in filamentous fungi, was the first step in the majority of studies related to *Aspergillus* species identification [16,17]. As shown in Table 2, the PCR sequencing of the ITS region identified *A. flavus* and *A. oryzae* with the same coefficient ratio (99/55%) and failed to discriminate between the two closely related species. The effectiveness of *CaM* in discriminating between *Aspergillus niger* and *Aspergillus tubingensis*, which belong to the *Aspergillus* Section *Nigri* was reported [18]. *CaM* has also been reported as an efficient marker in distinguishing between *A. fumigatus* and *Aspergillus lentulus*, two closely related species [19]. In our study, the partial sequences of the *CaM* gene showed that, with the exception of the two isolates (MZ291491 and MZ291495), eight isolates were identified as *A. flavus*, with an identification rate lower than 100%. Therefore, an easy-to-use, accurate, rapid, and cost-effective approach for differentiating the two species was still needed. Our results highlight that the *cyp51A* gene can be considered as a potential discriminative genetic marker in the differentiation between *A. flavus* and *A. oryzae*. The *cyp51* gene plays a proven role in the synthesis of ergosterol and is used as a key element in studies of azole resistance mechanisms in various fungi [20]. In fungi, mutations in *cyp51* cause resistance to azoles, a class of drugs that are used to treat invasive mycoses in mammals and in plants. This gene demonstrated highly conserved characteristic motifs, but very low overall sequence similarities; thus, it may be a promising target with which to find nucleotide variations [21,22,23]. The use of the *cyp51* gene in studies focused on fungal diversity is uncommon; however, our primary evaluation revealed that this gene could be reliable and helpful in the discrimination between *A. flavus* and *A. oryzae*. To assess the *cyp51A* gene’s ability in species identification, we first performed a qualitative PCR assay, which revealed that the *cyp51A* gene can separate *A. flavus* and *A. oryzae* from other aspergilli. Although *A. flavus* and *A. oryzae* produced sharp specific bands on electrophoresis panels, *A. fumigatus* and *A. niger* did not (Figure 3).

The *cyp51A* gene’s PCR products were also sequenced for each of the ten isolates. It demonstrated a novel and valuable performance in the field of evolutionary divergence under its own name, fully distinguishing between two very similar species, *A. flavus* and *A. oryzae*, with a 100% identification score (Table 2). A closer inspection of a partial *cyp51A* gene sequence alignment (1542 ntds) from ten isolates of *A. flavus* and *A. oryzae*, along with two standard isolates of both species, detected SNPs in four positions (Figure 1A). These findings provide significant value to support the strong genetic homology of *A. flavus* and *A. oryzae*.

## 5. Conclusions

Based on our findings, we suggest that the *cyp51A* gene is a promising target for the differentiation of *A. flavus* from *A. oryzae*. One limitation of the current study was that we only tested a small number of isolates from Iran. To rule out the effects of geographical variation, a larger set of isolates need to be tested before a solid conclusion can be made. The examination of a list of clinical and environmental isolates from different regions of the world mainly tropical and sub-tropical areas could be a useful addition to the publicly available databases.

## Figures and Tables

**Figure 1 pathogens-10-01279-f001:**
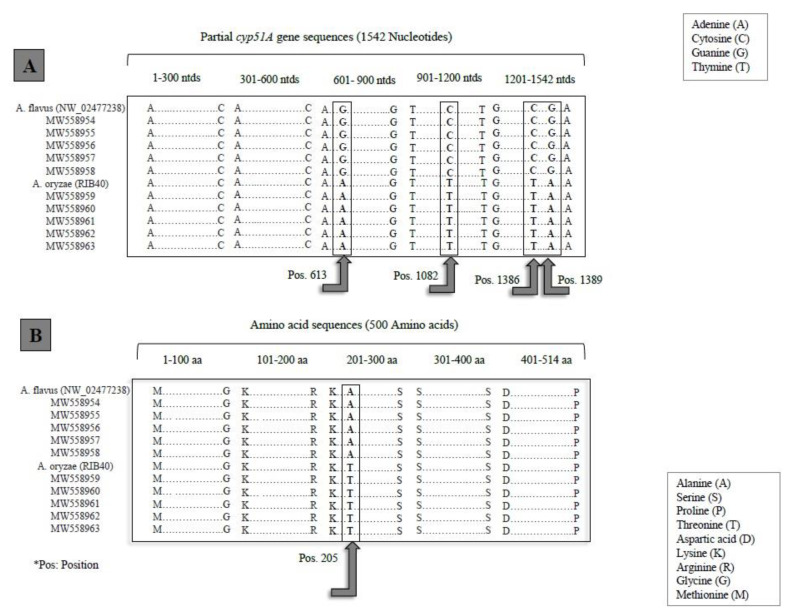
Multiple alignment of DNA sequences (**A**) and amino acid sequences (**B**) of *cyp51A* gene of *A. flavus* and *A. oryzae*.

**Figure 2 pathogens-10-01279-f002:**
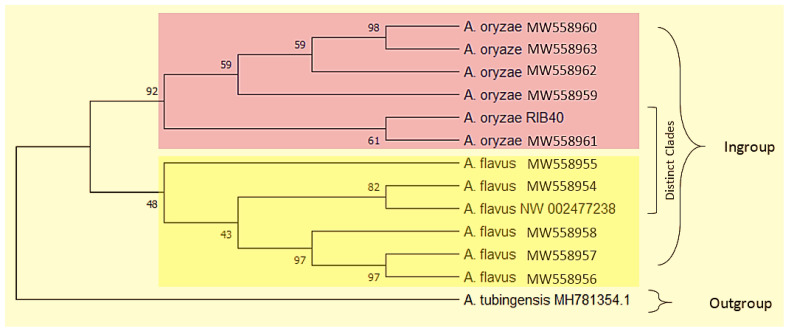
Phylogenetic tree of *A. flavus* and *A. oryzae* used in this study. *A. tubingensis* was used as the outgroup strain.

**Figure 3 pathogens-10-01279-f003:**
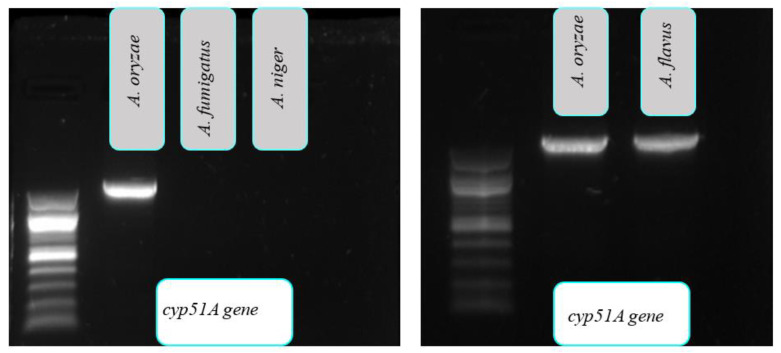
Gel electrophoresis of *cyp51A* gene PCR products.

**Table 1 pathogens-10-01279-t001:** Thermocycler program for PCR reaction of ITS, *benA*, *CaM*, and *cyp51A*.

Phase	Primer	Temperature	Time	
Initial Denaturation	ITS	94 °C	5′	
	*benA*	95 °C	5′	
	*CaM*	94 °C	5′	
	*cyp51A*	94 °C	7′	
Denaturation	ITS	95 °C	30″	
	*benA*	94 °C	45″	
	*CaM*	95 °C	30″	
	*cyp51A*	94 °C	30″	
Annealing	ITS	52 °C	60″	
	*benA*	60 °C	45″	
	*CaM*	60 °C	60″	
	*cyp51A*	52 °C	30″	
Extension	ITS	72 °C	2′	
	*benA*	72 °C	1′	
	*CaM*	72 °C	1′	
	*cyp51A*	72 °C	2′	
Final Extension	ITS	72 °C	7′	
	*benA*	72 °C	6′	
	*CaM*	72 °C	7′	
	*cyp51A*	72 °C	7′	
Cycle (Χ)	ITS	*benA*	*CaM*	*cyp51A*
	28	35	30	35

**Table 2 pathogens-10-01279-t002:** Identification rate of ITS, *BenA*, *CaM* and *cyp51A* genes. SP: sputum; BAL: Bronchoalveolar lavage.

Sample Code	Source	*BenA* Gene (Accession Number)	*ITS* (Accession Number)	*CaM* Gene (Accession Number)	*cyp51A* Gene (Accession Number)
1	BAL	100% *A. flavus* (MZ305275)	99.55% *A. flavus*, 99.55% *A. oryzae* (MZ055458)	99.48% *A. flavus*, 99.48% *A. oryzae* (MZ291495)	100% *A. flavus* (MW558954)
2	BAL	100% *A. flavus*, 100% *A. oryzae* (MZ305264)	99.55% *A. flavus*, 99.55% *A. oryzae* (MZ055461)	100% *A. flavus* (MZ291482)	100% *A. flavus* (MW558955)
3	SP	100% *A. flavus*, 100% *A. oryzae* (MZ305267)	99.55% *A. flavus*, 99.55% *A. oryzae* (MZ055465)	99.28% *A. flavus* (MZ291484)	100% *A. flavus* (MW558956)
4	BAL	100% *A. flavus* (MZ305269)	99.55% *A. flavus*, 99.55% *A. oryzae* (MZ055460)	99.61% A. *flavus* (MZ291489)	100% *A. flavus* (MW558957)
5	BAL	100% *A. flavus*, 100% *A. oryzae* (MZ305271)	99.55% *A. flavus*, 99.55% *A. oryzae* (MZ055468)	99.65% A. *flavus*, 99.65% *A. oryzae* (MZ291491)	100% *A. flavus* (MW558958)
6	SP	100% *A. flavus*, 100% *A. oryzae* (MZ305262)	99.55% *A. flavus*, 99.55% *A. oryzae* (MZ055459)	98.99% *A. flavus* (MZ291480)	100% *A. oryzae* (MW558959)
7	BAL	100% *A. flavus*, 100% *A. oryzae* (MZ305265)	99.55% *A. flavus*, 99.55% *A. oryzae* (MZ055462)	99.65% *A. flavus* (MZ291483)	100% *A. oryzae* (MW558960)
8	SP	100% *A. flavus*, 100% *A. oryzae* (MZ305263)	99.55% *A. flavus*, 99.55% *A. oryzae* (MZ055466)	96.93% *A. flavus* (MZ291487)	100% *A. oryzae* (MW558961)
9	BAL	100% *A. flavus* (MZ305273)	99.55% *A. flavus*, 99.55% *A. oryzae* (MZ055467)	100% *A. flavus* (MZ291493)	100% *A. oryzae* (MW558962)
10	SP	100% *A. flavus* (MZ305272)	99.55% *A. flavus*, 99.55% *A. oryzae* (MZ055469)	97.82% *A. flavus* (MZ291492)	100% *A. oryzae* (MW558963)

## Data Availability

The data presented in this study are available on request from the corresponding authors.

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
