# Peer review of "Differentiation of Aspergillus flavus from Aspergillus oryzae Targeting the cyp51A Gene"

_pathogens, 2021, doi:10.3390/pathogens10101279_

Round 1
Reviewer 1 Report
The authors described the molecular taxonomy of Aspergillus flavus and A. oryzae using cyp51A as a target gene in this manuscript. Even though the experiments seem adequate, it is not clear to me that there were any specific significance of cyp51A. There are minor but consistent grammatical errors throughout the manuscript that need to be addressed. Thus, I believe the manuscript is acceptable in “pathogens” after some major modifications.
- Please describe the reason why the authors selected cyp51A as a target gene. How about other toxin related genes, such as aflA?
- As you mentioned in Line 47-49, MALDI-TOF is not proper tool. Why you try MALDI-TOF analysis in this work? And where are resulting data (even though negative results)?
- All fungal names and gene names revised as Italic (Figures and Manuscript).
Reviewer 2 Report
The manuscript presented for review is very well written. contains a current problem based on current knowledge. The presented results have an additional advantage of practical application in a diagnostic laboratory. I have included minor comments in the PDF file.

Author Response
Response: Thank you for the reviewer’s attention. We made modifications on the manuscript based on the reviewers’ comments and replied in pdf format (attached file) as reviewer asked.

Round 2
Reviewer 1 Report
I think that the authors did not fully respond my questions and suggestions.
- The descriptions about cyp51A were not contain in revised version.
- The gene names were not change such as benA.
- If the authors would like to not generate MALDI-TOF results, please delete "Materials and methods parts".
Author Response
Please see the attachment

This manuscript is a resubmission of an earlier submission. The following is a list of the peer review reports and author responses from that submission.
Round 1
Reviewer 1 Report
Nargesi et al. report the posible use of CYP51A sequencing as taxonomic tool for differentiate A. oryzae from A. flavus. The manuscript is well written but I think that more strains shoud be added to make the conclusions stronger. Some cryptic species of the flavi section should be added to the study for sure, knowing that cryptic species of other sections show CYP51A sequence differences (e.g. A. lentulus CYP51A/A. fumigatus sensu stricto CYP51A).
Minor points to consider
Line 65: It should be species instead of strains. Am I ritgth?
Line 70: The origen of the used standards strains (oryzae (RIB40) and A. flavus (NRRL3357)) should be mentioned and how do the original identification was performed.
Line 81 to 89: Primers: Which sequences were used as template? The genebank accesion numbers are needed specially for CYP51A. It is not clear if authors use an universal CYP51A primer pair or one designed using the A. oryzae sequence or the A. flavus sequences. References are needed for all the other used primers.
Line 90: Trade mark of the enzyme kit is missing. Remember that someone would need it to replicate the experiments.
Line 93: It is not correct to use one sequencing primer despite the sequence lenght. Both directions are needed to confirm that any nt. change is not due to a sequencing error. Moreover, I am asuming that authors use amplifications primers as sequencing primers. However, that point should be stated.
Line 155: Title should be changed to something like ". Comparison of discriminatory potency between genetic markers". We already know which markers are being compared.
Table 2: genbank numbers should be in parenthesis (as it is stated in the first line of the table)
I do not think that figure 4 is usefull.
Discusion:
Figure 6 and data about it should be placed in the results section
Reviewer 2 Report
The authors showed a clue to differentiate between Aspergillus flavus and A. oryzae using cyp51A gene. Although the results shown in this study are clear, they have no novelty. In my view, the authors cited an article by Kjærbølling (Reference number 2), therefore, they know that the closest relative of A. oryzae is not A. flavus, but A. minisclerotigenes or A. aflatoxiformans. However, they did not include those species in the manuscript. The authors need to include these species data and should discuss the relationship between A. flavus and A. oryzae. Furthermore, they used only 10 clinical isolates. And the reason why the isolates were selected from 200 A. flavus strains. The authors described that they were A. flavus, but were they not A. flavus as shown in lines 59 and 60? (The author wrote 10 isolates showed >99% similarity to beta-tubulin gene of A. flavus/A. oryzae. Were 190 strains <99% similarity? And were they A. flavus or not?) The sample number is very small. The authors need to analyze more strains and more sequence data obtained in public. At the last, they showed colony morphology in Figure 1. Which strain is in each photo? Why did not you prepare the same presentation: A is forming three giant colonies, while B is just forming colonies.
Reviewer 3 Report
Dear authors,
this is an interesting work. You must have done a lot of work before coming across cyp51A.
I like the idea of presenting PCR thermal profiles for 4 genes. However, you have unnecessarily placed Figure 6 in the discussion rather than the results.
I found no justification in the text for why the distinction between the two species is important, useful or necessary. There was mention of different treatments, is that correct? If so, I suggest that this aspect be elaborated on.
L45 - delete one "in"
L 101 - delete the space between "bootstraps" and "."
L 106 - add "." after (MS)
L139 - add Latin name in italics and space after green before (B)
L153 - add "." after (B)
L210 - add "." at the end of the sentence, same in L222 and in Fig. 5.
in discussionL11 - remove extra space and check rest of the manuscript (l26, 34, 43, 64...)
L45 - restate sentence and do not start with cyp51.
L49 - "." cannot start a line